# Adaptation and Measurement Invariance by Gender of the Flourishing Scale in a Colombian Sample

**DOI:** 10.3390/ijerph18052664

**Published:** 2021-03-06

**Authors:** Marta Martín-Carbonell, Begoña Espejo, Irene Checa, Martha Fernández-Daza

**Affiliations:** 1Psychology Department, Cooperative University of Colombia, Troncal del Caribe S/N, 470002 Santa Marta, Colombia; martha.fernandez@campusucc.edu.co; 2Department of Behavioral Sciences Methodology, University of Valencia, Av. Blasco Ibáñez, 21, 46010 Valencia, Spain; bespejo@uv.es (B.E.); irene.checa@uv.es (I.C.)

**Keywords:** flourishing scale, wellbeing, structural equation modeling, psychometric properties, measurement invariance, confirmatory factor analysis, Colombian population, health, quality of life, psychological assessment

## Abstract

There is increasing interest in the study of flourishing as an indicator of subjective wellbeing. The objective herein was to adapt and study the psychometric properties of Diener’s Flourishing Scale (FS) among the Colombian population. Accordingly, a cross-sectional study was conducted with a non-probability sample of 1255 Colombian adults. The scale’s structure, invariance by gender, and convergent and concurrent validity were studied from a confirmatory perspective using structural equation models. The confirmatory factor analysis showed excellent fit indicators for the one-dimensional structure (CFI = 0.985, RMSEA = 0.039, SRMR = 0.020) as well as for the convergent (CFI = 0.909, RMSEA = 0.050, SRMR = 0.063) and concurrent (CFI = 0.966, RMSEA = 0.036, SRMR = 0.041) validity models. The correlations calculated among flourishing with positive and negative effects (PANAS), satisfaction with life (SWL), and optimism and pessimism (LOT) were statistically significant and as expected. Configural, metric, and scalar invariance across gender was confirmed. Percentiles were provided for the total score. The FS scale was a valid and reliable measure to assess high levels of wellbeing among the Colombian population.

## 1. Introduction

According to the cross-sectional, longitudinal, and experimental studies conducted, subjective wellbeing has been shown to be associated with a wide range of positive outcomes such as happiness, health, and a longer life [1,2], which explains the importance that this area of study has gained in the last fifty years.

Research in the past few decades shows two main traditions: one related to happiness (hedonic) and the other associated with the development of human potential (eudemonic), which considers wellbeing to be more related to an ability to effectively manage the environment, a sense of purpose and meaning, and a feeling of personal growth. The concept of flourishing appears in this second meaning, which has received particular attention, as demonstrated by different authors that have been key in developing this concept with their own peculiarities [1,3,4,5].

There is a consensus that flourishing refers to the “experience that life is going well” [1]. People with a high level of flourishing are vital and function positively in both the private and social aspects. They are willing to develop, improve, and expand their potential as a person and are able to build warm and trusting relationships with others [6]. “Flourishing” people are also more likely to better enjoy social relationships and experience fewer limitations in their daily activities [1], and they contribute to their communities [5,7]. The term is considered to be synonymous with a high level of mental wellbeing as well as an indicator of mental health [1,3].

Currently, there are several scales that can be used to study this construct, such as the Mental Health Continuum (MHC-LF) [3], the Mental Health Continuum-Short Form (MHC-SF) [4] and the Multidimensional Flourishing Scale [5]. However, the most researched scale may be the Flourishing Scale (FS) [6], which has quickly become popular because of its simple application and interpretation and its powerful conceptual and empirical basis. One of its advantages is that it produces data that can be easily interpreted by a wide range of potential end-users working in clinical, regulatory, and health promotion contexts among the population. It is a brief scale, and the author has made it freely available for use [7].

The FS was designed to measure social–psychological prosperity from a person’s own point of view [6], as other authors [8,9] found that, based on hypotheses from the humanist approach, there are universal psychological needs such as the need for competence, affiliation, and self-acceptance. They also considered its relationship with social capital [3], which is related to an interest and commitment to contributing to the wellbeing of others, which gives life meaning and purpose.

Thus, the FS includes several items related to having support and satisfying relationships, contributing to the happiness of others, and being respected by others. It also includes an item on having a life with purpose and meaning, and one on being engaged and interested in activities, as well as questions related to self-respect, optimism, and feeling competent and capable in activities that are important to the person.

The original FS comprised 12 items, but it was later reduced to 8. In the study of this latest version of 8 items, carried out with a sample of 689 participants, the FS showed good psychometric properties, with a Cronbach’s alpha of 0.87 and good test–retest reliability (r = 0.71).

The FS has been validated in different continents and many countries, such as Oceania (New Zealand [10]) and Europe (Portugal [11]; Germany [12]; France [13]; Italy [14]; Spain [15]; Greece [16]; Turkey [17] and Russia [18]). In Asia, we found studies from Japan [19], China [20], Iran [21], and India [22]. In Latin America, there have been communicated studies in Brazil [23], Puerto Rico [24], and Peru [25].

All these studies have consistently found a one-factor structure explaining between 45% (for example, Brazil) and 73.1% (for example, Japan) of the variance. In addition, most of the validations conducted confirmatory factor analyses, showing good fit scores (comparative fit index (CFI) between 0.971 and 0.986; root-mean-square error of approximation (RMSEA) between 0.041 and 0.08). Regarding reliability, all validation studies showed internal consistency scores between 0.83 and 0.95.

Convergent validity has been tested using instruments including the Subjective Happiness Scale [26], Satisfaction with Life Scale [27], Revised Life Orientation Test (R-LOT] [28], Positive and Negative Affect Schedule [21], Chinese Virtues Questionnaire [20], and Brief Symptom Inventory [20]. Convergent validity correlations with the above-mentioned scales ranged from 0.28 to 0.67, all of them being statistically significant. Some studies mention the testing of discriminant validity using the Hopkins Symptoms Checklist [29], Perceived Stress Scale (PSS) [19], and the Center for Epidemiological Studies Depression Scale [10].

There are several Spanish adaptations of the FS developed for Spain. In addition to the adaptation carried out by Checa, Perales, and Espejo [15], one by by de la Fuente, Parra, and Sánchez-Queija [30]. There are also two Spanish versions, one from Puerto Rico [24] and the other from Peru [25]. However, some of them do not have a back translation or do not describe the procedure used for translation. We found another adaptation of the FS with samples from Colombia and Spain [31], but this study has some limitations: first, the authors did not use a reverse translation. On the other hand, they used a five-point Likert scale, instead of the seven-point Likert scale proposed in the original questionnaire and did not give an explanation for this. Another problem is that the authors only evaluated metric invariance, but semantic, idiomatic, experiential, and conceptual equivalence is required in a cross-cultural adaptation [32], not just metric equivalence. In addition, a sample of university students was used for the analyses, convergent and criterion validity was not studied, and the same sample was used to perform the exploratory and confirmatory factor analysis [33].

The International Test Commission (ITC) recently issued new guidelines on the adaptation of tests from one culture to another, which include “offering relevant empirical information on the equivalence of the construct, equivalence of the method, and equivalence between the items in all populations involved”, “collecting information and evidence on the reliability and validity of the adapted version of the test among the populations involved”, and “establishing the level of comparability between scores from different populations using a data analysis or appropriate alignment designs” [34,35]. Although these guidelines are well known, they are not applied as frequently as would be ideal [34].

The present study took these recommendations into account and aimed to address the limitations of the previous Spanish validation of the FS among the Colombian population. For these reasons, this work aimed to adapt the Spanish version of the FS questionnaire [15] to the Colombian population and study its psychometric properties in a large and general sample. On the one hand, the factor structure has been studied through a confirmatory factor analysis. Another objective was the test of gender-based measurement invariance. The third aim was to study convergent and concurrent validity using different subjective wellbeing measures. Structural equation modeling methodology has been used to study the factor structure of the FS, measurement invariance, and convergent and concurrent validity.

## 2. Methods

### 2.1. Procedure

The first step was to adapt the version of Checa et al. [15] to the Spanish of Colombia. Following the recommendations by Muñiz et al. [32], an initial qualitative pilot study was conducted. The pilot study’s participants were selected using purposeful theoretical sampling serially until obtaining data saturation. In total, 14 people were included based on their willingness to collaborate and after ensuring they were Colombian adults (nine women and five men), with different education levels (eight people with a university education, three with high school diplomas, and three with primary school education), and between the ages of 18 and 81. The scale was applied using paper and pencil, and in an online version. The analysis of the participants’ responses revealed that the wording of the items in the version for Spaniards was appropriate for the Colombian context and that the participants correctly understood the items in both versions. The Colombian version of the Flourishing Scale is shown in the Appendix A (Table A1).

Participants were invited to participate through different means (email, social networks, and face-to-face). For the online data collection, the Limesurvey platform was used. In all cases, we explained the objective of the research and gave them the link to access. When accessing the survey, an explanation of the study was presented. Subsequently, before answering the survey, participants had to read and accept the informed consent.

The data were part of a broader study that aimed to study the scales’ psychometric properties to assess subjective wellbeing. The Ethics Committee of the Cooperative University of Colombia approved the study to guarantee the confidentiality and anonymity of the data, in accordance with the Colombian Law of Data.

### 2.2. Participants

A non-probabilistic sample of 1255 participants was used. Participants were included if they identified themselves as Colombian and were adults (18 years of age or older). The average age of the sample is 25.62 years (*SD* = 8.60, Minimum 18, Maximum 67). The majority were women, single, were married or cohabiting, had college or high-school academic studies and regarding the employment situation, the majority were students. In Table 1 is shown a detailed description of the sample, including frequencies (*N*) and percentages (%).

### 2.3. Measures

#### 2.3.1. Flourishing Scale (FS)

The Flourishing Scale is composed of eight items that describe positive human functioning and assesses positive relationships, feelings of competence, as well as meaning and purpose in life [11]. The items are answered with a 7-point Likert-type scale that ranges from 1 (strongly disagree) to 7 (strongly agree). Total scores can range from 8 (strong disagreement with all the items) to 56 (strong agreement with all the items), so higher scores indicate that people perceived themselves with a very positive image. This questionnaire has been adapted in a general sample of Spaniards, showing good psychometric properties [21]. For this study, this Spanish version was used because the Spanish adaptation process was conducted using the International Test Commission (ITC) criteria [36,37].

#### 2.3.2. Life Orientation Test-Revised (LOT-R)

This questionnaire measures optimism and pessimism [38,39]. This scale is made up of 10 items. Of them, only three items measure optimism, and another three items measure pessimism. The other four items are control items. The answer scale is a 5-point Likert-type scale that ranges from 1 (strongly disagree) to 5 (strongly agree). The total score for the subscale can range from 3 to 15, so the higher scores on each subscale indicate high optimism or high pessimism, respectively. The LOT-R has been adapted to the Colombian context showing good psychometric properties [40,41]. In this sample, Chronbach’s alpha for the Optimism subscale is 0.693 and for the Pessimism subscale 0.636.

#### 2.3.3. Satisfaction with Life Scale (SWLS)

The Satisfaction with Life Scale is a general measure of satisfaction with the quality of life perceived [27]. It is composed of five items that are answered with a 7-point Likert-type scale that ranges from 1 (strongly disagree) to 7 (strongly agree), so higher scores indicate higher satisfaction. This questionnaire has been adapted to the Colombian context using a general sample and has shown good psychometric properties [42]. Chronbach’s alpha in the present sample is 0.842.

#### 2.3.4. Positive and Negative Affect Scale (PANAS)

This questionnaire measures positive and negative affects [43]. It is composed of 10 positive items and 10 negative items. All of them are answered with a 5-point Likert-type scale that ranges from 1 (not at all) to 5 (extremely), so total scores in each subscale can range from 10 to 50. The respondents rate the frequency of their feelings during the last four weeks, and higher scores indicate higher positive or negative affect, respectively. The Colombian version used herein has been preliminarily validated for Colombian women showing good psychometric properties [44]. In the sample of this study, Chronbach’s alpha was 0.814 for the positive affect subscale, and 0.885 for negative affect subscale.

### 2.4. Data Analysis

As there is an important theoretical basis regarding the one-factor structure of the questionnaire, an exploratory factor analysis has not been performed. To study the factor structure of the FS, one confirmatory factor analysis (CFA) was estimated to test the one-dimensional model. Owing to the sensitivity to the sample size of the χ^2^ goodness of fit test, different indexes have been used to determine model fit: the comparative fit index (CFI), the Tucker–Lewis index (TLI), the root-mean-square error of approximation (RMSEA), and the standardized root-mean-square residual (SRMR). Values of 0.90 for the CFI and the TLI, as well as values between 0.06 and 0.08 for the RMSEA indicate acceptable model fit. Values above 0.95 for the CFI and the TLI, and values below 0.05 for the RMSEA indicate good fit to the model [36,37,45].To study reliability, we have used the Composite Reliability Index (CRI) and the Average Variance Extracted Index (AVEI). Values above 0.70 for the AVEI are considered good, and values of 0.50 are considered acceptable. For the CRI, values above 0.70 are considered good [46]. All values outside this range were considered not acceptable.

The gender-based configural, metric and scalar measurement invariance has been studied, too. With a sample size greater than 300, a change of less than −0.010 in CFI, complemented by a change of less than 0.015 in RMSEA or a change of less than 0.030 in SRMR, would indicate that there is invariance [47]. To study convergent and concurrent validity of the FS, two structural equation models were specified. In these models, the items and their underlying factors of the wellbeing measures considered have been included. Convergent validity was studied by including the PANAS-P and PANAS-N items with the FS items. Concurrent validity was studied in the same way but using the other measures mentioned above: optimism, pessimism and satisfaction with life.

All these data analyses (CFA, measurement invariance, and convergent and concurrent validity) were conducted using Mplus 8.4 [48]. Maximum likelihood robust (MLR) estimation was used to estimate the parameters. Observed data have been considered ordinal, but some authors suggest that MLR estimation can be used in CFA models with a non-normal distribution of data if the number of response categories for the items is greater than four [49,50]. In this case, the variability in the parameter estimates is relatively small and MLR offers less biased standard error estimates, as well as good estimates of the correlations between the factors [51]. On the other hand, as the response scale of the items was considered ordinal, to estimate the homogeneity indices of the items, corrected item-total poliserial correlations were calculated [52] as indicators of corrected homogeneity indices [53]. Finally, to describe sociodemographic variables, obtain descriptive statistics of FS, and evaluate gender-based differences, IBM SPSS 26 was used [54].

## 3. Results

In Table 2 are shown the descriptive data of the FS scale items with mean, standard deviation, skewness, kurtosis, and item-total corrected polyserial correlations.

One CFA was carried out to test the one-dimensional structure of FS, showing excellent goodness of fit: CFI = 0.985, TLI = 0.979, RMSEA = 0.039, RMSEA 90% interval = [0.028, −0.051], and SRMR = 0.020. All factorial loadings were statistically significant (*p* < 0.001) and ranged between 0.611 and 0.841. The AVEI (0.758) and the CRI (0.906) were excellent. Item-total corrected polyserial correlations showed very good values and were statistically significant, ranging from 0.596 to 0.766.

As can be seen in Table 3, both the estimated model to study convergent validity, and the estimated model to study concurrent validity, showed a very good fit to the data. All coefficients of the model were statistically significant (*p* < 0.001). The correlation coefficients among latent variables are displayed in Table 4.

In Table 5 are shown the results for the measurement invariance models by gender. The results indicate good fit of the one-factor model for men and women, and also, the FS showed invariance by gender. As these results let us compare means by gender, after fixing latent mean values to zero for males, no differences for gender were observed (*b* = 0.094, *z* = 1.552, *p* = 0.121).

In Table 6 descriptive statistics for the total score in the scale in this sample. As no statistically significant differences were found between the observed means for men (mean = 5.79) and women (mean = 5.89) (t (1252) = −1.357, *p* = 0.175).

## 4. Discussion

The purpose of this study consisted of adapting and studying the psychometric properties of the Spanish version of the FS following administration in a Colombian sample.

Adequate reliability and validity was observed. The CFA provided support for a one-dimensional flourishing construct using an eight-item scale. These outcomes are consistent with multiple studies in different countries [55], in the original study [6], in Peru [25], and in Spain [15]. In turn, the similarities of the results in all the scale’s validations in different countries suggest that the scale does not seem to be greatly affected by variables and cultural attributes. At a theoretical level, our data confirmed the consistency of the scale with that proposed by its authors, thus, this suggests a certain theoretical consistency regarding the psychological processes that facilitate and develop flourishing and human wellbeing, which is in line with the theoretical foundations of the psychology of wellbeing and positive psychology.

Furthermore, the one-factor structure of the FS showed scalar invariance (equal factor structure, factor loadings, and intercepts) across gender, supporting the results that have been found previously in other countries [10,11,12,13,14,15,16,17,18,19,20,21,22,23,24,25]. A lack of scalar invariance would mean that men and women are thinking about different constructs when we measure wellbeing, leading to a validity problem that would not allow for a comparison between both groups or for both groups to be studied together.

Moreover, our study includes the confirmatory reliability estimate, Average Variance Extracted Index (AVEI) and Composite Reliability Index (CRI), as well as the check for gender-based measurement invariance, which was previously only studied in the Greece version [16]. These results are important because a hypothesis of the generalization of a construct to other cultures has to be evaluated according to each situation [56].

Excellent evidence of convergent validity was found with PANAS as well as of concurrent validity using the LOT-R and SWLS variables. It is important to underline that we study convergent and concurrent validity of the FS, using an SEM. In this way, a much more reliable estimate of the relationship between the measurements is obtained. These outcomes are consistent with multiple studies in different countries [55] in the original study [6], in Peru [25], and in Spain [15]. In this sense, our data provide evidence for the importance of the flourishing construct in determining subjective wellbeing and for the validity of this scale in terms of its measurement.

In this study, high scores have been found in the FS items, which has also occurred in previous studies, such as in the study of the original scale [6] and also in the study with a Spanish sample of the general population [15]. However, because FS measures high levels of well-being [3], it is reasonable to think that, in a continuum of psychological wellbeing, flourishing would be located on the more positive side.

So, the tendency to experience high levels of wellbeing was confirmed, as the scores of the Colombian sample showed a non-normal distribution and bias toward high scores, which is expected because this measure is an indicator related to the most positive aspects of wellbeing. The values found are higher than those reported by Diener, which indicates cultural differences. According to the world database on happiness administered by the Erasmus University of Rotterdam, which includes 90 countries, Colombia’s inhabitants feel the happiest of all countries [56], but this view can conceal serious social problems and behaviors that can be harmful in the medium term if “happiness” or “wellbeing” is confused with joy [57]. Various studies have been conducted with the Colombian population, which were consistent in finding notable differences regarding wellbeing. This can be attributed to socioeconomic, regional, sociodemographic, and personal factors [58,59,60], among others.

This adaptation has overcome many of the limitations of a recent Spanish translation validated in Colombia and Spain [31] by taking into account the ITC recommendations for the adaptation of tests and the checklist of the compliance criteria that was recently proposed by Hernández et al. [34]. First, we selected the Spanish versions of the items, which were developed with the back-translation procedure, as is recommended [35]. However, as the objective was to achieve a linguistically correct and culturally adapted test that would measure the psychological construct with precision and validity, using an appropriate language for the population to be evaluated, current recommendations regarding the verification of adaptation were considered, which state that the adaptation verification required an iterative debugging process to arrive at a final consensual version. Therefore, a qualitative pilot study was conducted to provide evidence that both the test instructions and the item content had a similar meaning for Colombians and to assess the equivalence of the test administration mode (i.e., paper and pencil versus computer-controlled).

### Limitations and Future Directions

One limitation is the type of non-probability sampling used according to the availability, which limits the generalization of results, considering Colombia’s cultural diversity. The sample of this study shows differences in the proportion in which age is distributed in the Colombian population, according to data from the latest National Population and Housing Census of Colombia. In our sample there is a higher proportion of young adults (82% were under 30 years old and less than 1% of the subjects were over 65 years old), while in Colombia those over 65 are 9.1% and the population between 18 and 30 years old is approximately 16%. The sample of the present study has a higher educational level and does not include any illiterate person, although 5.9% of Colombians cannot read or write. These characteristics of the sample are probably due to the strategy used to obtain the information. In addition, administering the test online restricted participants by only allowing people with access to the Internet to participate. For these reasons, future studies with larger and more representative samples will be necessary to establish score scales.

Additionally, for future studies, examining temporal reliability with the test–retest technique is recommended. In turn, the scale’s psychometric properties should be investigated in other populations, such as the rural population and those of the other regions of Colombia. Conducting a comparison analysis based on other demographic variables such as age, socioeconomic status, and academic background would also be enriching. Research into cross-cultural invariance is recommended to better understand the cultural, socioeconomic, and demographic determining factors of flourishing.

In Colombia, studies on the association of psychosocial and environmental factors are scarce, but they are considered necessary due to the context of violence, social insecurity, and labor and economic instability in the country [61], as well as the accelerated deterioration of the environment, especially in the Colombian Caribbean. A recent meta-analysis [62] concluded that there is a strong and positive relationship between people’s pro-environmental behaviors and subjective well-being, and initial evidence that this relationship may be stronger the more clearly the meaning is reflected by behaviors and subjective well-being indicators. In this sense, flourishing could possibly be a useful indicator for policy makers and sustainability programs with positive impact on both people and the environment.

## 5. Conclusions

The unifactorial structure of the version of the FS questionnaire was verified in the Colombian sample, as well as the gender-invariance and its convergent and concurrent validity with other habitual measures of well-being, such as the Scale of Positive and Negative Affects (PANAS), the optimism and satisfaction with life. It is recommended that researchers and users interested in using the scale in Colombia take into account the limitations of this study.

## Figures and Tables

**Table 1 ijerph-18-02664-t001:** Sociodemographic characteristics of the sample.

		*N*	%
Gender	Women	806	64.2
	Men	449	35.8
Personal situation	Single	948	75.5
	Married or cohabiting	276	22
	Divorced	27	2.2
	Widowed	4	0.3
Educational level	Primary school studies	38	3
	Secondary school studies	162	12.9
	High school studies	517	41.2
	College studies	445	35.5
	Undergraduate studies	93	7.4
Main activity	Studying	878	70
	Working	298	23.7
	Unemployed, inactive or retired	79	6.3

**Table 2 ijerph-18-02664-t002:** Statistics of the items of the Flourishing Scale and corrected item-total polyserial correlations.

Item	M	*SD*	Sk	Kt	IT	SE
1	6.09	1.47	−2.14	4.25	0.659	0.014
2	5.41	1.45	−1.17	1.08	0.596	0.013
3	5.85	1.37	−1.77	3.23	0.759	0.010
4	6.10	1.29	−2.21	5.44	0.766	0.011
5	5.69	1.28	−1.52	2.75	0.693	0.013
6	5.93	1.24	−1.79	3.93	0.759	0.009
7	6.00	1.38	−1.79	3.17	0.757	0.010
8	5.77	1.25	−1.44	2.49	0.685	0.009

Note: M = mean; *SD* = standard deviation; Sk = skewness; Kt = kurtosis; IT = item-total corrected polyserial correlations; SE = standard error for the item-total corrected correlations.

**Table 3 ijerph-18-02664-t003:** Models to study the validity of the Flourishing Scale.

Model	χ²	df	CFI	TLI	RMSEA	RMSEA 90% CI	SRMR
Convergent validity	1415.669 *	347	0.909	0.901	0.050	0.047, 0.052	0.063
Concurrent validity	348.411 *	146	0.966	0.960	0.036	0.032, 0.040	0.041

Note: **χ² =** Chi square; df = degrees of freedom; CFI = comparative fit index; TLI = Tucker–Lewis index; RMSEA = root-mean-square error of approximation; 90% CI = 90% confidence interval; SRMR = standardized root-mean-squared residual. * *p* < 0.001.

**Table 4 ijerph-18-02664-t004:** Correlation coefficients (standard errors) between the latent variables of the Flourishing Scale and wellbeing measures.

Convergent Validity			Concurrent Validity			
	FS	PANAS-P		FS	OPT	PES
PANAS-P	0.461 (0.036) *	-	OPT	0.588 (0.043) *	-	
PANAS-N	−0.211 (0.036) *	−0.238 (0.042) *	PES	−0.186 (0.039) *	−0.261 (0.048) *	-
			SWLS	0.577 (0.038) *	0.736 (0.029) *	−0.242 (0.040) *

Note: FS = Flourishing Scale; PANAS-P = PANAS (Positive Subscale); PANAS-N = PANAS (Negative Subscale); OPT = Life Orientation Test (Optimism Subscale); PES = Life Orientation Test (Pessimism Subscale); SWLS = Satisfaction with Life Scale. * *p* < 0.001.

**Table 5 ijerph-18-02664-t005:** Measurement invariance models of the Flourishing Scale by gender (reference group: men).

Model	χ²	df	Δχ²	Δgl	CFI	RMSEA	SRMR	ΔCFI	ΔRMSEA	ΔSRMR
Men	19.946	20			1.00	0.000	0.020			
Women	58.397 *	20			0.978	0.049	0.024			
Configural	79.778 *	40	-	-	0.985	0.040	0.023	-	-	-
Metric	92.552 *	47	12.301	7	0.983	0.039	0.048	-0.002	−0.001	0.025
Scalar	103.181 *	54	8.120	7	0.981	0.038	0.046	-0.002	−0.001	−0.002

Note: df = degrees of freedom; Δχ² = Chi Square increase; Δgl = increase in degrees of freedom; CFI = comparative fit index; RMSEA = root-mean-square error of approximation; SRMR = standardized root mean square residual; ΔCFI = CFI increase; ΔRMSEA = RMSEA increase; ΔSRMR = SRMR increase. * *p* < 0.001.

**Table 6 ijerph-18-02664-t006:** Descriptive statistics and Flourishing Scale (FS) norms for the total score (percentile rankings).

Statistics		
Flourishing Scale Total Score		
Mean		46.790
Median		49
Mode		48
Standard deviation		8.635
Skewness		−1.988
Standard error of skewness		0.069
Kurtosis		5.066
Standard error of Kurtosis		0.138
Minimum		7
Maximum		56
Percentiles	5	31.0
	10	35.6
	15	39.0
	20	42.0
	25	44.0
	30	46.0
	35	47.0
	40	48.0
	45	48.0
	50	49.0
	55	50.0
	60	50.0
	65	51.0
	70	52.0
	75	52.0
	80	53.0
	85	54.0
	90	55.0
	95	56.0

## Data Availability

Not applicable.

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
