# Peer review of "Adaptation and Measurement Invariance by Gender of the Flourishing Scale in a Colombian Sample"

_ijerph, 2021, doi:10.3390/ijerph18052664_

Round 1
Reviewer 1 Report
thanks for the opportunity ti review this manuscript,
I believe that the publication has been adequately improved, in particular as regards a better clarity in the methods.
Reviewer 2 Report
Thank you very much for taking into account my comments and suggestions. I hope your article will be well-received by the scientific community.
Reviewer 3 Report
My comments had been considered by authors, and the problems I mentioned had been corrected.
This manuscript is a resubmission of an earlier submission. The following is a list of the peer review reports and author responses from that submission.
Round 1
Reviewer 1 Report
This is an interesting study which, in my opinion,has some shaky points: first , non probabilistic sampling may have contributed to a distortion of the data . I personally appreciate the probabilistic method.
A second point that i would re-evaluate is a case and control approach that would make the study more robust.
Reviewer 2 Report
- The authors used different sampling means. The number by mean should be described.
- Table 5 might be preferable to a figure (scatter gram).
Reviewer 3 Report
GENERAL COMMENTS
Thank you for the opportunity to review your paper. Although the topic of the paper might be interesting to psychologists and other social scientists, I feel that it does not fit the scope of the journal which is to focus “... on the publication of scientific and technical information on the impacts of natural phenomena and anthropogenic factors on the quality of our environment, the interrelationships between environmental health and the quality of life, as well as the socio-cultural, political, economic, and legal considerations related to environmental stewardship, environmental medicine, and public health”.
SPECIFIC COMMENTS
P1, L33: Period is needed.
P4, L162: It might be important to explain what is the rationale behind the sample size selection. Have you conduced a power analysis? Also, another important component of this analysis would be to mention if your sample size reflects the demographic characteristics of the Colombian population. In other words, is the proportion of sexes, as well as education and employment levels similar to what someone is expected to find in Colombia?
P5, L204-2013: Please consider indicating what was considered as “not acceptable”. For instance, you can say: “all values outside this range were considered not acceptable”.
P8, Discussion: I feel the discussion and conclusion sections should be expanded to discuss the use of the findings during exposure to different natural phenomena (i.e., how could a wider audience leverage your findings during exposure to different natural phenomenal, in keeping it with the theme of the journal)?
Reviewer 4 Report
Regarding the Title
The study was not used a a General Colombian Sample – it is a misleading title.
Regarding the Introduction:
The concept if the flourishing is explained in the first tree paragraph with 7 references. I am not sure that a shorter introduction would not be more proper.
The measurement possibilities and the advantages of the FS scale are summarized in the next paragraph.
The function of the next paragraph is not clear (Line 55-60). The term of social capital is mentioned between quotation marks, in spite the fact that this term is used commonly, therefore the quotation marks are not necessary; the sentence should be reformulated:
1 - comparing flourishing with social capital as it was defined originally
2 - emphasizing social component of the flourishing
The next paragraph (Line 61-65) seems to be a repetition of the former paragraphs content. I am not sure that it is necessary. (There is no reference in this paragraph which shows that authors think also that there is no new information provided by these sentences.)
The basic properties of the original FS with 12 items are summarized in the Line 66-73 paragraph. The N= 689 is not explained in the text. (Number of subject used in the cited study?)
The next three paragraphs are about the international adaptations’ experiences and convergent validity. (Line 74-84)
In the paragraph of the line 92-104 the Spanish versions are mentioned with listing the problems with their adaptation. The following paragraph introduces the methodological standard of the cultural adaptation of a test.
Regarding the Objectives:
The line 112-114 sentence and the last paragraph of the Introductions summarizes the aims. They should be compile into a distinct paragraph.
Objectives are well defined:
- to adapt the Spanish version of the FS questionnaire to the Colombian population,
- to determine the factor structure,
- to test gender- invariance,
- to study convergent and concurrent validity.
The paragraph should be more focused to the objectives:
- The “Thus, FS has shown good psychometric properties in multiple studies and is widely used in different well-being studies and in many different countries.” sentence is not necessary in this section. Its content is properly inserted into the former paragraphs.
- Sentences of “A lack of scalar invariance would mean that men and women are thinking about different constructs when we measure wellbeing, leading to a validity problem that would not allow for a comparison between both groups or for both groups to be studied together.” and “Structural equation modeling methodology has been used to study the factor structure of the SPANE, measurement invariance, and convergent and concurrent validity. Since there is an important theoretical basis regarding the one-factor structure of the questionnaire, an exploratory factor analysis has not been performed.” are not about the aims, these should be inserted into the discussion and into the methodological sections.
Altogether the content of the Introduction is proper but it should be much more focused. It needs rewriting.
Regarding the Methods:
I think that the title should be corrected: there are no materials used in the study.
Procedure:
- Selection of the version developed by Checa et al has been described. Without describing the argumentation behind the mentioned consensus, this section is not informative. Readers cannot evaluate how established was the decision. Focusing to the objectives this paragraph is not required.
- The initial qualitative pilot with 14 subjects on the paper-and-pencil and on the on-line version is described by many sentences which are not able to describe the argumentation for the decision (the original Spanish items are proper for Colombian setting). This section (two paragraphs) should be reduced.
- The paragraph of line 153-156 is about the recruitment of the main study? If the aim was the Colombian adaptation of FS then the target population is Colombian people. Here, the sampling frame HAS to be presented. (If the sampling could not ensure the representativeness then the Colombian adaptation CANNOT be achieved, and the title of the paper should be corrected; the real aim of the paper should be reformulate…)
Participants:
The relationship between the sample and the target population HAS to be reported.
Measures:
Flourishing Scale:
The 8-item FS is used. In the introduction, the properties of the 12-item is described. (The use of the shorter version is not mentioned in the section on aims.) It is not clear WHY to describe details on the longer FS in the introduction and use the shorter one.
The abbreviation of ITC for the International Test Commission should be introduced when it is mentioned first.
Data Analysis:
The description of the methods applied are summarized properly.
The usual threshold values of the model fitting for confirmatory factor analysis (Line 208-213) are not necessary to be reported.
Regarding the Results:
Item specific descriptive statistics are proper.
I miss the inclusion of the text of the item. Readers should understand better the statistical results having the (English versions) of the questions.
CFA, convergent and concurrent validity, and gender invariability are properly described.
The reported “preliminary norms of the Spanish version of the FS were calculated for the Colombian population” is important. But it is simply the description of the observed FS scores (descriptive result) and not the population level norms – if the sample is representative of the Colombian population then this table is about the useful standards, if the sample is not representative then the “preliminary norms” are misleading.
Valid and Lost parts are not necessary in the Table 5. Similarly, the minimum, maximum, and percentiles are normal numbers, therefore decimals should not be reported.
Regarding the Discussion:
The “standard scores for the Colombian population” in line 312 is overinterpretation of the results.
Comparisons to other study’s’ results are insufficient. Data on comparisons are no reported at all.
Limitations and Future Directions:
Authors acknowledge the problem of non-representativeness of the sample studied WITHOUT ANALYZING THE CONSEQUENCES of this validity problem.
Regarding the Conclusions:
The Conclusions should answer the study questions raised in the Objectives section.
The first paragraph is a Discussion, not about conclusions.
The second paragraph is about conclusion drawn from the reported results. The usefulness, concluded by authors, is the function of the representativeness of the studied sample and the ITC-based evaluation. Since the former requirement is (seemingly) not met, only the second one does it, the conclusion is not established.